# Plant-Based Burgers in the Spotlight: A Detailed Composition Evaluation and Comprehensive Discussion on Nutrient Adequacy

**DOI:** 10.3390/foods14030372

**Published:** 2025-01-23

**Authors:** Katia Regina Biazotto, Ana Carolina Hadlich Xavier, Rosane Ribeiro de Mattos, Júnior Mendes Furlan, Roger Wagner, Daniel Henrique Bandoni, Veridiana Vera de Rosso

**Affiliations:** 1Food Labeling Observatory, Nutrition and Food Service Research Center (CPPNAC), Federal University of São Paulo (UNIFESP), Santos 11015-020, SP, Brazil; 2Graduate Program in Nutrition, Federal University of São Paulo (UNIFESP), São Paulo 04023-062, SP, Brazil; 3Chromatography and Food Analysis Research Group, Federal University of Pampa (UNIPAMPA), Itaqui 97650-000, RS, Brazil; anacarolinahadlich5@gmail.com (A.C.H.X.);; 4Department of Technology and Food Science, Federal University of Santa Maria (UFSM), Santa Maria 97105-900, RS, Brazil; rosanefmattos@gmail.com (R.R.d.M.); juniorfurlan@unipampa.edu.br (J.M.F.)

**Keywords:** plant-based burgers, protein, fatty acids, essential amino acids

## Abstract

This study aimed to evaluate the nutritional profile of commercial plant-based burgers (PBBs) available in Brazil and to assess their suitability for fulfilling nutritional requirements. Seven PBBs were selected, based on the different protein sources used in the formulations. The proximate and mineral compositions were evaluated, and the fatty acid and amino acid profiles were determined. The protein contents ranged from 5.25 ± 0.37 to 13.55 ± 1.16 g/100 g in the PBBs made from quinoa and a mix of proteins, respectively. The studied PBBs can offer between 46 and 71% of the essential amino acid (EAA) requirements. In addition, the total amount of EAAs provided the nutritional requirements established by the FAO/UN for all of the PBBs, considering the 100 g portion intake. The total fat content in the PBBs ranged from 3.51 ± 0.11 to 12.74 ± 1.93 g/100 g. Linoleic acid and oleic acid were the major fatty acids in the three PBBs, while myristic acid was the major fatty acid in one PBB studied. This study revealed significant differences in the nutritional composition between PBBs marketed in Brazil. Additionally, the lack of regulation allows for considerable variation in their nutritional profiles, making it difficult to compare them with those of meat burgers.

## 1. Introduction

In the current half of this century, the world’s population will grow to approximately 9 billion; in the future, the global demand for food, feed, and fibers will nearly double [1]. As the population grows, the diversity in dietary needs and preferences necessitates a broader range of food products. Increasing food production should not only focus on quantity but also on the nutritional quality of food, to combat malnutrition and related health issues.

Plant-based diets align more with meeting sustainable development goals (SDGs) [1]. In addition to promoting food security to achieve zero hunger (SDG 2), they are often associated with numerous health benefits, including a lower risk of heart disease, hypertension, type 2 diabetes, and certain cancers [2,3]. Improved health outcomes can improve well-being and reduce healthcare costs (SDG 3). Thus, the demand for plant-based meat analogs (PBMAs) has surged over the past decade, driven by a confluence of factors related to health, environmental sustainability, animal welfare, and shifting consumer preferences [4,5,6,7]. PBMAs offer an alternative for those who want to reduce or exclude meat consumption without abandoning the taste and texture of meat [8].

The plant-based food market has evolved beyond traditional options such as veggie burgers and tofu [9]. Currently, there are a wide array of plant-based products that mimic the taste and texture of animal-derived foods. The market is diversifying to cater to various dietary preferences and cultural tastes. In the 2010s, the first plant-based meat analog was launched in the American market and promoted disruption in the food sector [10]. This trend has gained worldwide recognition, and the global market for plant-based meat analog production reached USD 6.1 billion by 2022 [11]. Projections from the main market analysis agencies indicate that plant-based consumption will skyrocket to 10% of the total meat consumption, or USD 140 billion globally, by 2029 [12]. This market growth is expected to be driven by Gen Z and millennials (aged 16–40 years), with 66% anticipating an increase in their consumption of similar plant-based products over the next 10 years [11]. Plant-based meat analogs, particularly plant-based burgers (PBBs), are widely consumed by the mainstream public. However, there is limited information available regarding the nutritional composition of these products compared with traditional meat products. Many consumers perceive plant-based products as either inherently healthy or overly processed, and nutritionally inadequate [13,14].

Recent studies have pointed out that one of the main limitations of the partial or total replacement of meat products with plant-based meat analogs is food essentialism [14]. Food essentialism is made up of beliefs that food categories have innate, immutable “essences” that are responsible for their properties [15]. Consumers often associate the nutritional value and naturalness of meat with meat products, whereas similar plant-based alternatives are frequently perceived as ultra-processed and unnatural [13,14]. Classifying PBBs as ultra-processed may contribute to stigmatizing their consumption, reinforcing the perception that they are less natural, nutritious, or healthy compared to animal-based products [16]. An effective strategy to address the effects of food essentialism on the replacement of meat products is to enhance knowledge about PBMAs. This includes understanding their characteristics, the functions of their ingredients, their nutritional composition, and the potential impact of their consumption on human health [13,14].

However, most existing studies have focused on assessing the nutritional quality of PBMAs based on the nutritional information provided on food labels [17,18,19,20,21,22,23,24]. Comprehensive studies on the nutritional composition of plant-based products using laboratory analytical approaches are scarce in the literature [25,26,27,28]. Furthermore, there is a clear need for more studies that include both qualitative and quantitative evaluations of fatty and amino acids. This is currently lacking, which significantly impedes the accurate assessment of the nutritional quality of these foods.

While plant-based ingredients, in their raw form, often lack some of the nutrients found in animal products, modern PBMAs can be specifically designed to address these gaps. Technological advancements in food science and growing consumer demand for nutritionally comparable substitutes have led to the development of fortified and enhanced PBMAs. This enrichment helps make PBMAs more nutritionally complete and competitive with animal-based products. In addition, the range of ingredients used in plant-based burgers reflects manufacturers’ attempts to recreate the sensory and nutritional characteristics of traditional meat [12].

Legumes like soy and peas are considered high-quality plant proteins used in PBB fabrication, as they contain all nine essential amino acids in reasonable amounts [29]. However, other ingredients may not independently provide a complete protein profile, requiring either ingredient combinations or fortification to ensure that they meet the daily amino acid requirements. Soy and peas are particularly common, but ingredients like chickpeas, quinoa, and lentils are also frequently used. Therefore, plant-based burgers available on the market, which utilize a variety of cereals and legumes as protein sources combined in different proportions, require further investigation. Additionally, the sources of lipids in these products are diverse. Alongside widely available options such as soybean oil, alternatives like coconut, sunflower, and olive oils are also employed [30,31].

This study aimed to fill this gap by evaluating the detailed nutritional composition of plant-based burgers available on the Brazilian market. In addition, we discuss the implications of nutrient adequacy considering the Dietary Reference Intakes and fatty acid health indicators, as well as the contribution of the various ingredients used in product formulations. This study aims to provide evidence-based insights to dispel misconceptions and shed light on the nutritional strengths and limitations of plant-based burgers (PBBs). In doing so, this equips consumers with the knowledge needed to make informed dietary decisions.

## 2. Materials and Methods

### 2.1. Materials

All reagents and organic solvents used in proximate analysis were acquired from Synth (Diadema, São Paulo, Brazil). The Total Dietary Fiber Assay Kit (K-TDFR-200) was supplied by Megazyme Neogen (Lansing, MI, USA). All authentic standards were supplied by Sigma-Aldrich (St. Louis, MO, USA): 1. Fatty acid standards ((FAME Mix 37 (P/N 47885-U), cis/trans isomers of linoleic acid methyl esters (P/N 47791), a mixture of linolenic acid methyl esters (P/N 47792), transvacenic methyl ester (P/N 46905-U), and docosapentaenoic acid methyl ester (P/N 47563-U)); 2. amino acid standards (L-alanine, L-glycine, L-valine, L-leucine, L-isoleucine, DL-norleucine, L-proline, L-methionine, L-serine, L-threonine, L-phenylalanine, L-aspartic acid, hydroxyproline, l-cysteine, l-glutamic acid, L-asparagine, L-lysine, L-glutamine, L-cystine, L-arginine, L-histidine, L-tyrosine, and L-tryptophan); 3. mineral standards (Ca, Fe, Na, Mg, Mn, Zn, Cu, K, and P).

### 2.2. Sample Collection

Three criteria guided the selection of PBBs analyzed in this study: (1) wide availability in Brazilian markets, (2) popularity among consumers, and (3) diversity in protein ingredients used in their formulations. Three lots of each of the seven PBBs analyzed (*n* = 21) were purchased from local stores in São Paulo, São Paulo State, Brazil.

All samples were acquired in March 2023 and brought to the Bromatology Laboratory of the Federal University of São Paulo (Santos City, São Paulo State, Brazil) on ice. Each burger sample was vacuum-packed in plastic bags, frozen, and stored at −18 °C until analysis. Prior to analysis, all samples were thawed for 24 h at 8 °C and homogenized using an industrial processor. Table 1 shows the characterization of the samples according to protein and fat sources.

### 2.3. Proximate and Mineral Analysis

All analyses were performed according to the Official Methods of Analysis of the Association of Official Analytical Chemists [32]. The FAO/INFOODS Guidelines for Checking Food Composition Data [33] were used for data quality validation.

The moisture content was determined using the AOAC 925.09-1925 method [34] and the total ash content was measured using the AOAC 940.26 method [35]. Total nitrogen was determined employing the Kjeldahl method, according to AOAC 950.09 [36], and the nitrogen content was multiplied by a conversion factor (6.25) to estimate the protein content. Total fat was determined according to the Soxhlet method—AOAC 963.15 [37] and total dietary fiber was evaluated using an enzymatic–gravimetric method—AOAC 985.29 [38]. The available carbohydrate content was calculated as the difference, considering the values for moisture, protein, total fat, and dietary fiber. Data were expressed as the mean ± standard deviation from triplicate analysis.

Mineral profiles were determined after the mineralization of the samples using a microwave digestion system (employing concentrated nitric acid at 175 °C for 10 min). Quantification was carried out in inductively coupled plasma optical emission spectrometry (ICP-OES 720-ES, Varian, Palo Alto, CA, USA) employing wavelength specific to each mineral: sodium at 589.59 nm; potassium at 766.94 nm; calcium at 315.88 nm; magnesium at 285.21 nm; iron at 259.94 nm; manganese at 257.61 nm; zinc at 213.85 nm; phosphorus at 177.49 nm; and copper at 324.75 nm. The equipment operating conditions used were a 2 mL/min for sample aspiration rate, plasma power of 1 kW, coolant flow of 15 L/min, auxiliary gas flow of 1.5 L/min, nebulizer pressure of 200 kPa, axial mode for reading, and integration time of 20 s. The minerals were quantified using external calibration curves with a minimum of eight concentration levels: between 100 and 500 ppb for Fe, Na, Cu, Zn, Ca, Mg, and Mn and between 800 and 4000 ppb for P and K. Data are expressed as the mean ± standard deviation from quintuplicate analyses.

### 2.4. Determination of Fatty Acid Profile and Lipid Fraction Quality Evaluation

Lipids were extracted according to the method described by Hara and Radin [39]. Briefly, 3.0 g of the sample was mixed with 28 mL of hexane/isopropanol (3:2 plus 0.05% of BHT) in an Ultra Turrax homogenizer T18 (IKA, Shanghai, China). The samples were shaken for 1 h in an orbital shaker NT-145 (Novatecnica, Piracicaba, Brazil), and 12 mL of sodium sulfate (6.66%) was added, followed by stirring for another 10 min. The mixture was centrifuged (8000 rpm for 10 min) and the apolar fraction was collected. A 5 mL aliquot (≅25 mg of lipids) was obtained and mixed with 0.5 mL of C23:0 (internal standard = 1.012 mg). The solvent was evaporated (T < 40 °C) and fatty acid methyl esters were prepared according to Hartman and Lago [40] and analyzed using a gas chromatograph (Varian, Star, 3400, Lima, OH, USA) equipped with a flame ionization detector (GC-FID). Chromatography was performed using an HP-88 column (100 m × 0.25 mm i.d., 0.20 µm). The initial column temperature was 80 °C for 2 min and it was then programmed to increase to 175 °C at 10 °C/min, which was maintained for 10 min, followed by an increase at 4 °C/min until reaching 210 °C, which was maintained for 5 min, followed by a new increment at 5 °C/min until reaching 230 °C, and then the temperature was finally maintained for 5 min at 230 °C. The injector temperature was set at 270 °C and operated in the split mode (1:20). Hydrogen was used as the carrier gas at a pressure of 30 psi and the detector temperature was maintained at 250 °C.

Fatty acid identification was performed by comparing the retention times of the samples with those of the standards: FAME Mix 37 (P/N 47885-U), cis/trans isomers of linoleic acid methyl esters (P/N 47791), isomer mixtures of linolenic acid methyl esters (P/N 47792), transvacenic methyl ester (P/N 46905-U), and docosapentaenoic acid methyl ester (P/N 47563-U) (Sigma-Aldrich, USA). Fatty acids were quantified using an internal standard (C23:0) considering the correction factors from the chain size of GC-FID and ester acid conversion [41]. Data were expressed as the mean ± standard deviation from triplicate analysis in terms of mg of fatty acids per gram of lipids.

### 2.5. Determination of Amino Acid Profile and Quality Evaluation

The amino acid profile was evaluated according to the method described by Furlan et al. [42] Briefly, to release amino acids through the hydrolysis of plant-based burgers, a 20 mL Schlenk glass flask was employed. In this process, 100 mg of the sample was added to 10 mL of hydrochloric acid (6 mol/L) containing 1 g/L of phenol as the catalyst. Acid hydrolysis was then conducted under vacuum (<50 mm Hg). The hydrolysates or internal standard (DL-norleucine—10 μg/L) were diluted (1:100) with HCl (0.1 mol/L) and subjected to derivatization with N-*tert*-butyldimethylsilyl-*N*-methyltrifluoroacetamide (MTBSTFA). This mixture was left to react for an additional 120 min at 100 °C.

Chromatographic analysis was performed using a GC–MS system (QP2010-Plus, Shimadzu Corporation, Kyoto, Japan). Amino acids were separated on an NST-5MS capillary column (30 m × 0.25 mm; 0.25 µm film thickness; Nano Separation Technologies, São Carlos, SP, Brazil). The operating conditions were as follows: injector temperature of 285 °C with a 10:1 split; initial temperature of 100 °C for 1 min, followed by a ramp up to 220 °C at 20 °C/min, then an increase to 250 °C at 5 °C/min, and subsequent increase to 260 °C at 2 °C/min, and isothermal maintenance for 1 min. The temperature was further increased to 265 °C at a rate of 2 °C/min. Following this, the temperature was raised to 285 °C at a rate of 5 °C/min and isothermally maintained for 1 min. Finally, it reached 300 °C at a rate of 15 °C/min and was held for 2 min, totaling 29.5 min. Helium (99.9995%) was used as carrier gas at a constant linear velocity of 40 cm/s. The GC–MS interface and electron impact ionization source (EI at 70 eV) temperatures were 280 °C and 210 °C, respectively.

The MS-quadrupole was operated in the full-scan mode, enabling us to acquire the total ion chromatogram at a mass-to-charge ratio (*m*/*z*) ranging from 50 to 500 for the unequivocal identification of amino acids. Selected ions were used for amino acid quantification. Analytes were positively identified by comparing their retention times and mass spectra with those of 23 amino acid analytical standards (Sigma-Aldrich, USA). Quantification was performed by external calibration (0.2–60 µg/mL), with areas previously normalized by the internal standard. Data are expressed as the mean ± standard deviation from triplicate analysis in terms of mg of amino acids per gram of protein and mg of amino acids per 100 g of PBB.

### 2.6. Data Validation

The accuracy of the analytical procedures was evaluated using certified reference material (CRM) (NIST-SRM1548B, National Institute of Standards and Technology, Gaithersburg, MD, USA) of a typical diet. The following components were tested using the CRM: 1. proximate composition (total solids, total fat, protein, and total fiber); 2. mineral composition (Ca, Cu, Fe, Mg, Fe, Mn, P, K, Na, and Zn); 3. amino acids (alanine, arginine, aspartic acid, cystine, methionine, isoleucine, and valine); 4. fatty acids (lauric acid, myristic acid, palmitic acid, stearic acid, linoleic acid, total trans C18:1 and C18:2 fatty acids, total polyunsaturated fatty acids, and total saturated fatty acids).

### 2.7. Nutrient Adequacy Evaluation of the PBBs

The contents of individual minerals (Fe, Zn, K, Na, and Cu), protein, and dietary fiber in a 100 g portion of plant-based burgers were compared to the recommended dietary intake values for adults (18–50 years old) according to the Institute of Medicine [43]. For protein quality evaluation, the chemical score for each essential amino acid was calculated using the amino acid scoring pattern recommended for adults (>18 years) [44]. A comparison was made between the daily requirement of essential amino acids for adults and the amount of essential amino acids in a 100 g serving portion of PBBs.

For quality lipid fraction evaluation, the following are defined: 1. Σ SAT= sum of saturated fatty acids, which included C6:0, C8:0, C9:0, C10:0, C11:0, C12:0, C13:0, C14:0, C15:0, C16:0, C17:0, C18:0, C19:0, C20:0, C21:0, C22:0, C23:0, and C24:0; 2. Σ MUFA= sum of monounsaturated fatty acids including C16:1n9, C16:1n7, C17:1, C18:1n9t, C18:1n9, C22:1n6, and C24:1n6; 3. Σ PUFA= sum of polyunsaturated fatty acids, which included C18:2n6, C18:2n6t, C18:3 (C18:3n6 and C18:3n3), C20:2n6, C20:3 (C20:3n6 and C20:3n3), and C22:2n6; 4. Σ TRANS = sum of trans fatty acids including C18:1n9t and C18:2n6t; 5. Σ ω6 = sum of omega-6 fatty acids including C22:1n6, C24:1n6, C18:2n6, C18:2n6t, C18:3n6, C20:2n6, and C20:3n6; 6. Σ ω3 = sum of omega-3 fatty acids including C18:3n3 and C20:3n3; 7. Σ ω6/Σ ω3 = ratio between the sum of omega-6 fatty acids and the sum of omega-3 fatty acids; 8. Σ PUFA/Σ SAT = ratio between sum of polyunsaturated fatty acids and sum of saturated fatty acids; 9. the atherogenic index (IA) and thrombogenic index (TI) were calculated using the formulas proposed by Ulbricht and Southgate [45], as shown in Equations (1) and (2).AI = [(C12:0) + (4 × C14:0) + (C16:0)/(∑ω6) + (∑ω3) + (∑MUFA)](1)TI = (C14:0 + C16:0 + C18:0)/[(0.5 × ∑MUFA) + (0.5 × ∑ω6) + (3 × ∑ω3) + (∑ω3/∑ω6)](2)

### 2.8. Statistical Analysis

The data were distributed normally and presented as the mean ± standard deviation. Differences were detected using analysis of variance (ANOVA) followed by Tukey’s significant difference post hoc test, and differences were considered significant at *p* < 0.05. Statistical analysis was performed using the software Statistica 7.0 (StatSoft, Tulsa, OK, USA).

## 3. Results and Discussion

Seven plant-based burgers from different protein sources available on the Brazilian market were analyzed. The detailed nutritional composition, including the mineral, fatty acid, and amino acid contents, was thoroughly examined.

### 3.1. Proximate and Mineral Composition

The plant-based burgers analyzed were selected according to different protein ingredients, including cereal and legume flours, textured proteins, protein concentrates, and protein isolates, which were used individually or as part of a protein mix. Similarly, the use of oils and fats of various types has also been noted (Table 1). The PBB composition results did not indicate the uniformity of a specific food category. The results of the proximate and mineral composition analyses of the PBBs are presented in Table 2. Appendix A shows the determined levels and accuracy (%) calculated from the reference values for the typical diet (NIST-SRM1548B) employed as reference material.

The protein levels in the QUI and CHP samples were significantly lower than those of the other PBBs. In terms of the total fat content, SOY1 was significantly higher, whereas QUI and CHP had significantly lower total fat contents than those of the other PBBs. The fiber content was significantly lower in QUI and PLENT, whereas the available carbohydrates were significantly higher in QUI and CHP than in the other PBBs. The available carbohydrate content was determined by subtracting the other components of the sample. The results showed that PPB SPGL, SPCHP, SOY1, and SOY2 had levels close to zero, whereas QUI, PLENT, and CHP had higher levels. In QUI and CHP, the total carbohydrate content (including available carbohydrates and fibers) quantified in the plant-based burgers reached 30.15% and 29.41%, respectively. Consequently, carbohydrates, excluding water, were the predominant components of these products.

The protein content in the Brazilian plant-based burgers ranged from 5.25 ± 0.37 to 13.55 ± 1.16 g/100 g for the products derived from quinoa (QUI) and protein mix (soy, pea, and gluten—SPGL), respectively. In comparison, plant-based burgers sold in Italy had a higher protein content, ranging from 14.62 to 16.17 g/100 g in soy burgers to 14.38 to 21.07 g/100 g in pea burgers [46]. Another study found that plant-based burgers marketed in the European Union had a higher protein content than that of Brazilian PBBs (median: 18.01 g/100 g; 95% CI: 13.30–18.43 g/100 g) [26]. Similar trends were observed in US PBBs, with protein levels ranging from 17.2 ± 0.9 to 22.0 ± 4.1 g/100 g [47].

In Brazil, no legislation defines the identity profile and nutrient quality standards for plant-based burgers, particularly in terms of the minimum protein and maximum total fat levels. This contrasts with the regulations for meat burgers, which require a minimum protein content of 15% and maximum fat content of 25% [48]. Therefore, when comparing the plant-based burgers analyzed in this study with the legal requirements for beef burgers, we found that none of the analyzed products reached the required protein standards for meat burgers. In another study carried out in Brazil, the protein content in pea PBBs and meat burgers was higher, reaching 24.6 and 24.1 g/100 g, respectively [49]. The protein content in plant-based burgers and meat burgers marketed in the European Union was similar, with medians varying between 18.1 and 17.96 g/100 g, respectively [26]. The analyzed PBB supplied between 11.0% and 28.2% of the protein intake required for women and between 9.3% and 24.0% for men, based on the Recommended Dietary Allowance (RDA) for adults [43].

The total fat levels varied between 3.51 ± 0.82 and 12.74 ± 1.93 g/100 g in the Brazilian plant-based burgers, while in Italian plant-based burgers, the total fat levels were 10.03–10.21 g/100 g for soy burgers and 6.22–9.83 g/100 g for pea burgers [46]. Higher levels of total fat were reported by De Marchi et al. [26] (median: 11.10 g/100 g; 95% CI: 8.76–19.08 g/100 g) in the European Union PBB.

In terms of the total fat levels determined in the PBBs, all the products showed levels lower than the maximum fat content established in the Brazilian regulations for beef burgers. Similarly, Higuera et al. [49] reported that Brazilian meat burgers exhibited a higher total fat content (26.5 ± 0.4 g/100 g) compared to soy and pea burgers, which ranged between 15.3 ± 0.2 and 14.1 ± 0.4 g/100 g, respectively. For products marketed in the European Union, the total fat levels for PBBs and meat burgers were similar, with medians ranging between 11.10 g/100 g (95% IC: 8.76–19.08) and 12.51 g/100 g (95% IC: 7.98 and 20.33) [26].

Unlike meat burgers, PBBs contain fibers. The fiber levels in the analyzed PBBs ranged from 6.88 ± 0.31 to 10.62 ± 0.55 g/100 g, which, according to Brazilian legislation, can be considered as high in fiber [50,51]. According to the Dietary Reference Intakes, a 100 g portion of these products supplies between 27.5% and 42.4% of the fiber intake requirements for women and between 18.0% and 27.9% for men (adults aged 18–50 years) [43]. Insoluble fibers were predominant in all the plant-based samples compared to soluble fibers. In addition, the insoluble/soluble fiber (IF/SF) ratio varied among the different plant-based burgers. For CHP, the IF/SF ratio was 85%/15%, whereas for SOY1, it was 58%/42%. Evidence suggests that soluble fibers play a role in certain health effects, such as blood glucose attenuation and cholesterol lowering, while insoluble fibers play a role in other health effects, such as laxation [52]. The presence of fiber is directly linked to gastrointestinal, immune, and metabolic health associated with plant-based diets [52,53,54].

The literature on the role of dietary fiber in mineral bioavailability is imprecise. In vitro studies have revealed mineral binding or physical entrapment, and both animal and human studies have failed to show negative effects on mineral absorption [55]. The colonic fermentation of dietary fibers produces short-chain fatty acids and increases the proliferation of epithelial cells, which, in turn, increases the absorptive surface area and promotes mineral absorption.

The total mineral content in food is represented by the ash content, which generally serves as an estimate of the main micronutrients of inorganic origin present in foods. No significant differences were found in the ash content among the various PBB samples analyzed. It is recommended to individually quantify key minerals such as Fe, Ca, K, Na, Mg, and Zn to obtain a more accurate overview of the mineral content in foods. Sodium and potassium were detected at the highest concentrations in PBBs compared to other minerals. These results were expected, given their presence in raw materials and the fact that these two components are often added as salts in the food industry. However, the sodium levels are not considered high, as Brazilian legislation sets the limit for classifying food as high in sodium at 600 mg/100 g [50,51].

The PLENT sample had a higher Fe content than those of the other PBBs. The samples that contained soy in their formulation showed Fe levels varying between 2.09 and 3.86 mg/100 g. These values were similar in soy burgers marketed in the European Union (median 2.6 mg/100 g; CI 95% 2.39; 3.31 mg/100 g) [26] and in another study from Brazil (3.0 mg/100 g) [49]. Yet, they were lower than those quantified in soy PBBs marketed in the United Kingdom (31.25 mg/100 g wet basis) [56]. Surprisingly, the Fe levels were higher in the PBBs made from soy than in meat burgers, and their bioaccessibility was equivalent to that of meat burgers [49,56]. The Fe bioavailability in the plant-based burgers (made from soy) analyzed using the Caco-2 cell model after in vitro digestion was comparable to that of a meat burger [56].

Consistent with the findings of this study, Luz et al. [57] observed significant variability in the mineral content and its corresponding bioaccessibility among Brazilian commercial PBBs. These differences were attributed to the ingredients used, the impact of the food matrix, and the chemical forms of the elements, which influence their solubility under digestive tract conditions [57]. The iron content ranged from 4.64 mg to 12.9 mg in the PBBs made with chickpea and pea, respectively, while the Fe bioaccessibility varied between 16.9% and 51% [57].

The Zn content in the PBBs varied between 0.61 and 1.69 mg/100 g for the SOY2 and CHP samples, respectively. In another study conducted with PBBs from Brazil, the zinc levels were similar for plant-based burgers made from pea and soy (1.1 ± 0.01 and 1.2 ± 0.02 mg/100 g, respectively) [49]. Different levels of Zn have been reported, depending on the type of protein used in the product formulation. For example, the Zn content in the pea burgers varied between 2.09 and 2.33 mg/100 g, while in the soy burgers it varied between 2.9 and 4.8 mg/100 g [47]. The zinc content ranged between 2.39 and 3.47 mg/100 g for the pea PBBs and 1.23 to 1.43 mg/100 g in the chickpea PBBs [57].

In addition, different results have been reported in the literature comparing the zinc levels between plant-based burgers and meat burgers. In some studies, the Zn levels were higher in plant-based burgers [47], whereas in others, the Zn levels were higher in meat burgers [26,56]. Another study reported no significant differences [49]. The same variability was observed for the bioaccessibility and bioavailability of Zn. Latunde-Dada et al. [56] reported that the Zn bioaccessibility and Caco-2 uptake were significantly higher in meat burgers than in plant-based burgers, while Higuera et al. [49] and Luz et al. [57] reported the opposite regarding Zn bioaccessibility.

Consuming 100 g of the analyzed plant-based burgers would contribute between 23.0% and 127.7% of the recommended daily intake (RDI) of Fe, as provided by QUI and PLENT, respectively [43]. The plant-based burgers supplied a lower Zn content regarding the recommended daily intake, providing between 6.8 and 15.3% Zn in the SPCHP and CHP samples, respectively. All the PBBs supplied 100% of the RDI for Cu and K.

The variation in the mineral levels present in the PBBs, as well as their bioaccessibility and bioavailability, is related to the protein source (leguminous, cereal, mycoprotein, algal, and others) and the manufacturing of ingredients used to obtain them [26,47,49,56]. Additionally, other compounds present in raw vegetable materials, such as phytates and mineral antagonists, can inhibit mineral absorption. However, antinutritional factors can be reduced or eliminated during the processing of raw vegetable materials, such as grinding combined with heat treatment, to inactivate phytase [58]. In addition, antinutritional compounds, such as myo-inositol phosphates and total dietary fiber, did not adversely impact the mineral bioaccessibility in the PBBs marketed in Brazil [57].

Although many studies have expressed concerns about replacing omnivorous diets with plant-based diets, particularly regarding the adequate intake of Fe and Zn, recent research has shown that the serum Fe levels in various groups were not significantly affected [59,60]. However, the total serum zinc concentrations were reduced in vegans [61]. This gap can easily be addressed by fortifying plant-based burgers with these micronutrients.

### 3.2. Fatty Acid Composition

Fats and oils contribute to the sensory properties of plant-based burgers, such as juiciness, tenderness, mouthfeel, and flavor release. In addition, they can contribute to the physical and textural properties of a product, such as its hardness, elasticity, cohesiveness, and gumminess [62]. The fat ingredients used in meat analogs include canola oil, coconut oil, sunflower oil, corn oil, sesame oil, cocoa butter, and many other sources of vegetable and plant oils [63].

The individual quantification (Table 3) of the fatty acids in the plant-based burgers demonstrated that C18:2 n6 (linoleic acid) was the major fatty acid in the SPGL, QUI, and CHP samples, accounting for 38.89, 43.06%, and 55.31%, respectively. In the SPCHP and PLENT samples, the majority of the fatty acids were C18:1 n9 (oleic acid), comprising 46.39 and 42.47% of the total, respectively. In SOY1, the major fatty acid was C12:0 (lauric acid), which is characteristic of coconut oil. For SOY2, the fatty acid profile was constituted of an equivalent distribution of oleic acid (23.61%), linoleic acid (22.53%), and lauric acid (22.52%). Furthermore, saturated fat was significantly higher in SOY1 and SOY2 than in the other PBBs (Figure 1) and higher than in meat burgers (Table 4) [64].

Linoleic acid is a characteristic vegetable oil derived from soybeans and sunflowers [65]. Oleic acid is commonly found in olive and canola oils [66,67]. Linoleic acid, oleic acid, and lauric acid have been frequently reported in studies evaluating the fatty acid composition of plant-based burgers [26,27,28]. Additionally, product reformulations are common in PBBs. The use of different lipid sources can significantly affect the fatty acid profile and nutritional quality of the final product. Swing et al. [47] reported an increase in saturated fatty acids in the reformulation of a PBB sold in the US market (from 19.6 ± 2.6% to 43.7 ± 5.0%) and a decrease in another PBB (from 89.9 ± 14.8% to 44.1 ± 5.1%). On the other hand, in PBBs sold in Canada, monounsaturated fatty acids predominated, ranging from 47.76 ± 0.12% to 65.66 ± 0.16% [27]. A unique study that analyzed PBBs as a complete meal (bread + plant-based burger + condiments) observed that saturated fatty acids were present in a lower proportion (2.2 g of SFA/100 g in 12.0 g total fat/100 g) [28].

The quality of the PBB lipid fraction was assessed based on three indices calculated from the fatty acid profile of each sample. The atherogenic index (AI) and thrombogenic index (TI) were used to estimate the relationship between diet and coronary heart disease and were calculated according to Ulbricht and Southgate [45] and the ω6/ω3 fatty acid ratio. The atherogenic index varied between 0.11 and 5.43 for CHP and SOY1, respectively. For SPGL, the QUI and CHP AIs were lower than those for the meat burger, whereas that of SOY1 was 8.5 times higher. The TI varied between 0.26 and 4.23 for the CHP and SOY1 samples, respectively. All PBBs, except SOY1, showed a lower TI than that of the meat burger, and that of SOY1 was 2.3 times higher. In general, food atherogenicity is affected by the levels of C12:0 (lauric acid), C14:0 (myristic acid), and C16:0 (palmitic acid), especially because myristic acid is the most atherogenic, with approximately four times the cholesterol-raising potential of palmitic acid [45]. Long-chain saturated fatty acids (C14:0, C16:0, and C18:0) accelerate thrombus formation, whereas MUFAs and PUFAs do not.

In plant-based burgers marketed in Italy, the atherogenic index was greater (median: 1.47) than that of meat burgers (median: 0.77), while the thrombogenic index was lower (median: 0.60) than that of meat burgers (median: 1.66) [26]. The authors attributed these results to the presence of coconut oil in the formulations of the plant-based burgers, as observed in some plant-based burgers marketed in Brazil. Other studies that used plant-based burgers marketed in Canada showed opposite results [27]. The AI and TI of the plant-based burgers were lower than those of meat burgers because of poor saturated fatty acid levels, especially those of short-chain saturated fatty acids.

Several studies have suggested using the ω6/ω3 fatty acid ratio in foods and diets as an indicator of the risk of developing cardiovascular diseases (CVDs), cancers, and inflammatory diseases. The incidence of CVD increases with higher ω6/ω3 fatty acid ratios. For instance, countries with more than 450 CVD deaths per 100,000 inhabitants typically have ω6/ω3 fatty acid ratios greater than 10 [68]. In contrast, a ratio of approximately four was associated with a 70% decrease in the risk of mortality from all types of diseases. Different dietary recommendations for optimum ω6/ω3 fatty acid ratios for health benefits have been adopted in different countries. The Food and Agriculture Organization of the United Nations (FAO/UN) recommends a wide range of ω6/ω3 fatty acid ratios (5–10) [69]. In contrast, lower ratios (<5) have been shown to reduce the CVD risk [68].

The ω6/ω3 fatty acid ratios of SOY1, SPGL, and CHP varied between 105.69 and 35.97, respectively. However, SPCHP and QUI showed ω6/ω3 fatty acid ratios lower than 10, reaching 2.07 and 9.43, respectively. In general, the vegetable oils employed in PBMA formulations exhibit higher ω6/ω3 fatty acid ratios, and include soy oil (6.6), olive oil (7.0), and sunflower oil (80.0) [65,68,70]. In contrast, linseed and rapeseed oils had lower ω6/ω3 fatty acid ratios, varying between 0.23 and 2.6, respectively. Thus, dietary recommendations involve consuming PBMAs with seed oils that are high in oleic acid (C18:1 n9) and low in linoleic acid (C18:2 n6), which may further reduce n6 PUFA intake [71]. The use of canola oil in PBBs marketed in Canada resulted in an excellent ω6/ω3 fatty acid ratio, ranging from 2.45 ± 0.01 to 4.76 ± 0.03 [27]. Plant-based burger development using gelled emulsions with chia and hemp oil as fat source resulted in low-fat PBBs with a PUFA/SFA ratio greater than 4.5 and ω6/ω3 lower than 4.0 [72].

Saturated fat plays an important role in many foods, including plant-based burgers, owing to their technological and organoleptic properties. In contrast, excessive consumption is associated with several cardiovascular problems. The use of edible structured oils that display a healthier fatty acid profile is one of the most promising alternatives [73]. As such, the development of oleogels, bigels, and proteins applied in emulsions has been of great interest as a solution for replacing saturated fats [73,74]. In addition, the use of linseed and rapeseed oils (rich in omega-3) in the development of structured oils can help reduce the intake of omega-6 fatty acids, which is promoted by using soybean and sunflower oils in the composition of PBMAs [75,76].

### 3.3. Amino Acid Profile and Protein Quality

The amino acid profiles of the seven plant-based burgers, expressed in mg/g of protein, are presented in Table 5. Figure 2 shows the essential (EAA) and non-essential (NEAA) amino acid composition expressed in g/100 g of plant-based burger (PBB). The sum of the total amino acids (ΣEAA + ΣNEAA) compared with the total protein levels determined using the Kjeldahl method showed substantial concordance. All of the PBBs presented varying amino acid profiles owing to different protein sources. The EAA content of the plant-based burgers was compared to the amino acid scoring pattern for adults, and the results are shown in Table 6. Figure 3 presents the percentage of the daily requirement of essential amino acids provided by a 100 g portion of PBBs.

Glutamic acid and aspartic acid were the most abundant non-essential amino acids detected in the PBBs analyzed. Similarly, lysine and aromatic amino acids (AAA-phenylalanine + tyrosine) were found at higher concentrations than essential amino acids (Figure 2).

All of the PBBs, except for QUI, showed methionine + cysteine (sulfur amino acids) as being limiting, compared to the scoring pattern required for adults [44]. QUI showed the lowest compliance with the requirement patterns for all essential amino acids (Figure 3). Additionally, leucine was also limiting in QUI (chemical score of 0.93), supplying only 9.89% of the daily requirement for this amino acid. The main protein ingredient of QUI is quinoa; thus, these results were surprising, since quinoa is a cereal with excellent nutritional value, particularly in terms of its amino acid profile. However, leucine, a limiting amino acid, has been previously reported in germplasms and planting locations [77]. In contrast, quinoa grown in Bolivia and Argentina also exhibited limited aromatic amino acids, as cysteine was not detected in any samples, and the amount of methionine present was insufficient to provide the requirement of 22 mg/g of protein [78]. This finding is important because it demonstrates the variations in the amino acid profile within the same cereal species.

The SOY1 sample showed a limiting score for valine (0.96), leucine (0.80), and sulfur amino acids (0.60). In addition, for these amino acids, one serving portion of 100 g of the burger provided 30.21% of the daily requirement for valine, 25.26% for leucine, and 18.09% for sulfur amino acids (Figure 3). According to the literature, soy presents sulfur amino acids as limiting [29,79]. However, the limiting supply of valine and leucine depends on the type of soy protein used; for example, isolated and concentrated soy proteins do not present limitations in these essential amino acids [79].

The SPGL, SPCHP, and PLENT PBBs developed formulations using a mixture of proteins. SPGL is composed of soy protein, pea protein, and gluten; SPCHP is composed of textured soy protein, pea protein, and chickpea flour; and PLENT is made up of textured pea protein, pea protein isolate, rice protein isolate, and lentil protein concentrate. As expected, these burgers showed sulfur amino acids as limiting, as legumes such as soybeans, chickpeas, lentils, and peas are limited in amino acids [29,80,81]. The CHP PBB had chickpeas as a protein source and, thus, also presented sulfur amino acids and leucine as limiting [80].

A study conducted on plant-based burgers (four brands: two soy-based and two pea-based) sold in Italy aimed to determine the protein quality compared to meat burgers (two brands of beef burgers). The two soy-based and pea-based burgers had the same limitation in terms of lysine [46]. This limitation was not observed in the other pea-based burger brands, leading the authors to speculate about the use of rice protein to address this issue. In another study conducted on PBBs marketed in the European Union, methionine was detected as the limiting amino acid in these products. In addition, cysteine and glutamic acid were more abundant in the PBBs than in meat burgers [26]. In both studies, the authors reported that the essential amino acid profile of the plant-based burgers was equivalent to that of the beef burgers. In addition, the meat burgers showed high levels of glycine and hydroxyproline due to the use of meat rich in connective tissue [26,46]. Although some EAAs were considered limiting in the studied PBBs, the total amount of essential amino acids provided the nutritional requirements established by the FAO/UN. For adults, this value is 291 mg EAA/g of protein [44]. The studied PBBs can offer between 46 and 71% more EAA than the requirements.

Usually, the first step in alternative meat manufacturing is protein extraction and functionalization. Proteins isolated from plants have been subjected to several processes to improve their functional characteristics. Following the formulation step, plant proteins are mixed with other ingredients to enhance the formation of a meaty texture and improve the taste [82]. The selection of protein raw materials should consider the digestible indispensable amino acid score (DIAAS). This index accounts for the composition of the EEA, standardized ileal digestibility, and reference pattern scoring of the EAA. For vegetable proteins, the DIAAS can range from 36 for corn to 100 for potato [83]. The higher the DIAAS (>75), the better the nutritional quality of the protein. A promising strategy for plant-based products to achieve a higher DIAAS is the use of protein mixtures. For example, soy/wheat/potato (25/20/55) and pea/wheat/soy (25/20/55) achieved DIAASs of 100 and 90, respectively [83]. In addition, legumes such as black beans, black-eyed beans, mung beans, peanuts, and chickpeas are promising due to their protein concentration and higher DIAAS [84].

In a recently published study involving 774 Brazilian individuals who followed a vegan diet, it was observed that while the majority of the caloric intake came from unprocessed foods (66.5%), the adequacy of the protein and essential amino acid intake depended on the consumption of processed and ultra-processed foods. The adequacy of the protein intake was 0.93; for essential amino acids, the adequacy ranged from 0.90 for lysine to 0.98 for aromatic amino acids [85]. These results demonstrated that the consumption of PBMAs can contribute to a plant-based diet with good nutritional quality. Recent studies have highlighted food essentialism as one of the key barriers to the partial or complete replacement of meat products with plant-based meat analogs [13,14]. Food essentialism is rooted in the belief that food categories possess inherent, unchanging “essences” that define and explain their shared properties [15]. Consumers often associate the nutritional value and naturalness of meat with meat products, whereas similar plant-based alternatives are frequently perceived as ultra-processed and unnatural [16]. Classifying PBBs as ultra-processed may contribute to stigmatizing their consumption, reinforcing the perception that they are less natural, nutritious, or healthy than animal-based products. In this context, the primary contribution of this study was to emphasize the nutritional quality of PBBs, particularly by highlighting the amino acid profile. This was achieved by determining the percentage of daily essential amino acid requirements supplied by 100 g of plant-based burgers.

The main limitation of this study lies in the need for a broader sampling of PBBs available in Brazil. While a previous study analyzing label information included 117 PBBs [22] with various protein and fat sources, this study focused on only 7 PBBs, which restricts the generalizability of the findings.

## 4. Conclusions

The variability in the nutritional composition observed in the PBBs analyzed in this study has also been corroborated by several other studies that have conducted bromatological analyses of PBBs. The lack of established identity and quality standards for PBMAs presents a significant challenge in evaluating the nutritional value and comparing the healthiness of these products to the foods they are designed to imitate.

One concerning nutritional aspect observed was that two of the analyzed PBBs contained higher levels of saturated fat than their meat analogs. Consequently, these two PBBs had the worst lipid quality markers (AI and TI) among those evaluated in this study. However, all of the PBBs had a total fat content lower than that found in meat burgers. The frequent use of vegetable oils in various formulations led to high levels of C18:2 n6 and, consequently, high ω6/ω3 fatty acid ratios in three of the studied PBBs.

All of the PBBs analyzed were considered to have a high dietary fiber content. In terms of the sodium content, none of the PBBs were considered high in sodium. Therefore, they are superior to their meat analogs, which are known for their high sodium content.

Although the protein content in all the analyzed PBBs was lower than the standards set for meat burgers, they provided the total essential amino acids recommended by the Dietary Reference Intakes. However, a limitation was found for sulfur amino acids (methionine and cysteine) in six of the seven analyzed PBBs. This result is common for proteins derived from legumes.

Based on the results of this study, it is crucial to highlight that a plant-based diet must meet the same nutritional criteria as other diets. Therefore, it should vary, consisting of foods from different groups such as cereals, tubers, legumes, and fruits, among others. Evaluating the nutritional composition of a single food group is insufficient for adequately assessing the health of any diet. Food scientists should make efforts to evaluate the impact of replacing meat and meat products using comprehensive approaches that focus not on individual foods but on diets with the partial and total replacement of meat and meat products.

## Figures and Tables

**Figure 1 foods-14-00372-f001:**
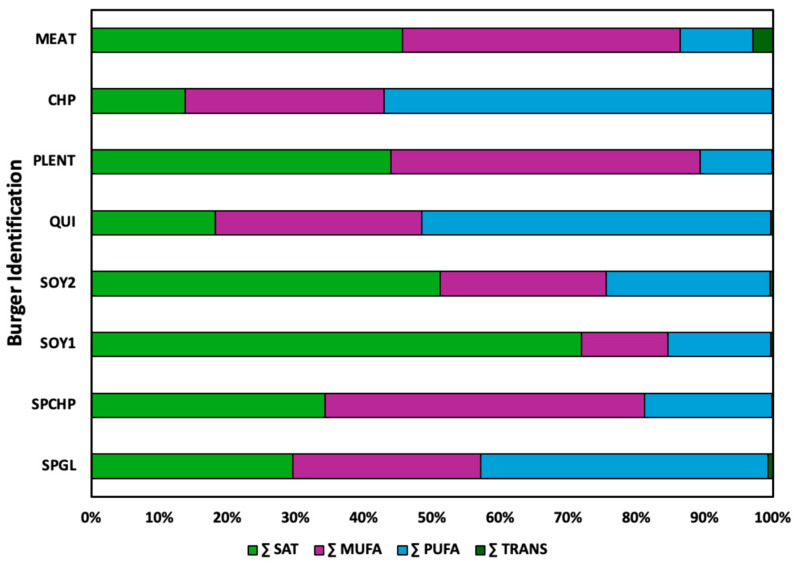
Percentage distribution of fatty acid composition in plant-based burgers. Σ SAT: sum of saturated fatty acids; Σ MUFA: sum of monounsaturated fatty acids; Σ PUFA: sum of polyunsaturated fatty acids; Σ TRANS: sum of *trans* fatty acids.

**Figure 2 foods-14-00372-f002:**
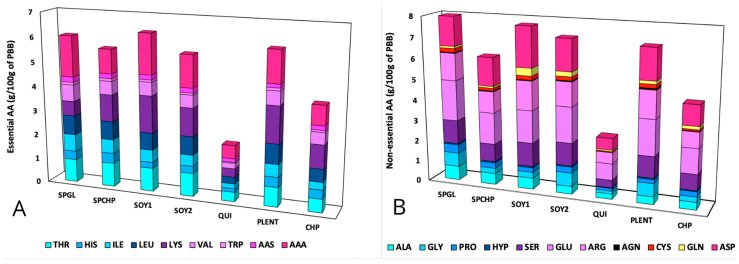
Amino acid composition of plant-based burgers expressed in mg/100 g. (**A**) Essential amino acids; (**B**) non-essential amino acids.

**Figure 3 foods-14-00372-f003:**
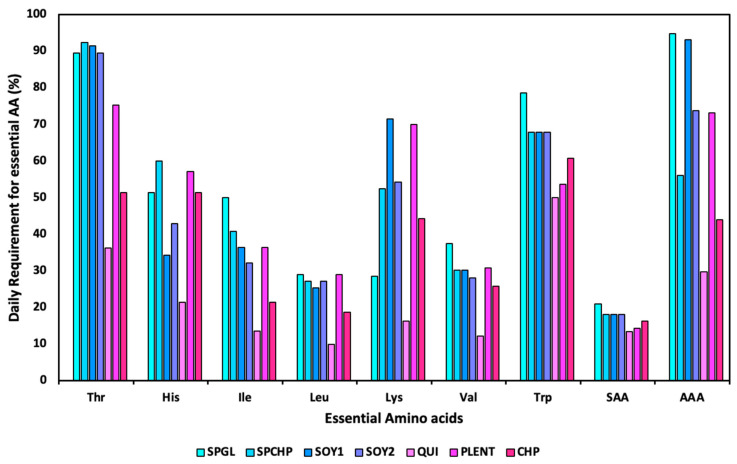
Daily percentage requirement of essential amino acids provided by 100 g of plant-based burgers. Thr: threonine; His: histidine; Ile: isoleucine; Leu: leucine; Lys: lysine; Val: valine; Trp: tryptophan; AAA: aromatic amino acids (phenylalanine + tyrosine); SAA: sulfur amino acids (methionine + cysteine).

**Table 1 foods-14-00372-t001:** Characteristics of plant-based burgers analyzed, according to protein and fat sources.

Sample	Code	Protein Source	Fat Source
1	SPGL	soy, pea, and gluten	vegetable oil and vegetable fat
2	SPCHP	soy, pea, and chickpea	coconut oil and vegetable fat
3	SOY1	Soy	vegetable fat and sunflower oil
4	SOY2	Soy	coconut oil and vegetable fat
5	QUI	white, red, and black quinoa	olive oil
6	PLENT	pea and lentil	sunflower oil and coconut oil
7	CHP	Chickpea	olive oil

**Table 2 foods-14-00372-t002:** Proximate and mineral composition of plant-based burgers marketed in Brazil.

	Plant-Based Burgers						
Component (g/100 g) *	SPGL	SPCHP	SOY1	SOY2	QUI	PLENT	CHP
Moisture	63.77 ± 2.42 ^A^	65.96 ± 0.30 ^A^	61.03 ± 4.41 ^A^	64.68 ± 4.05 ^A^	59.81 ± 0.30 ^A^	57.20 ± 2.24 ^B^	54.76 ± 1.21 ^B^
Ash	2.29 ± 2.15 ^A^	2.69 ± 0.36 ^A^	2.60 ± 0.27 ^A^	2.18 ± 0.48 ^A^	1.88 ± 0.10 ^A^	2.88 ± 0.32 ^A^	1.81 ± 0.08 ^A^
Proteins	13.55 ± 1.16 ^A^	12.79 ± 0.39 ^A^	12.46 ±1.60 ^A^	13.01 ± 1.29 ^A^	5.25 ± 0.37 ^B^	12.18 ± 0.49 ^A^	9.77 ± 0.35 ^C^
Total fat	9.61 ± 0.23 ^A^	9.90 ± 0.26 ^A^	12.74 ± 1.93 ^B^	8.72 ± 1.42 ^A^	3.65 ± 0.82 ^C,D^	10.26 ± 2.33 ^A^	3.51 ± 0.11 ^C,D^
Fibers	9.84 ± 0.76 ^A^	9.36 ± 0.25 ^A^	10.62 ± 0.55 ^B^	10.03 ± 1.02 ^A,B^	7.02 ± 0.34 ^C^	6.88 ± 0.31 ^C^	8.22 ± 0.52 ^D^
ACHO	0.00 ± 1.54 ^A^	0.00 ± 0.78 ^A^	0.00 ± 1.04 ^A^	1.61 ± 0.47 ^A^	22.34 ± 0.47 ^B^	10.44 ± 0.42 ^C^	21.39 ± 1.21 ^B^
Energy (Kcal/100 g)	180.00 ± 9.72 ^A^	177.70 ± 4.90 ^A,B^	206.99 ± 25.97 ^A^	177.08 ± 20.84 ^A,B^	150.23 ± 9.67 ^B^	186.34 ± 25.85 ^A^	189.11 ± 9.31 ^A^
Total	99.06 ± 1.67	100.70 ± 0.48	99.45 ± 1.39	100.23 ± 2.58	99.95 ± 0.28	99.84 ± 1.89	99.46 ± 0.98
**Mineral (mg/100 g) ****	**SPGL**	**SPCHP**	**SOY1**	**SOY2**	**QUI**	**PLENT**	**CHP**
Iron	3.86 ± 0.56 ^A^	3.58 ± 0.87 ^A^	2.09 ± 0.27 ^B^	2.21 ± 0.40 ^B^	1.84 ± 0.21 ^B^	10.22 ± 1.62 ^C^	2.95 ± 0.43 ^B^
Sodium	261.99 ± 27.19 ^A^	161.64 ± 20.85 ^B^	296.10 ± 24.00 ^A^	109.13 ± 179.14 ^B^	272.22 ± 47.25 ^A^	363.65 ± 121.56 ^A^	253.65 ± 66.31 ^A^
Potassium	248.65 ± 17.80 ^A^	431.81 ± 21.80 ^B^	452.08 ± 29.10 ^B^	321.61 ± 34.94 ^C^	202.52 ± 2.17 ^D^	368.96 ± 6.99 ^C^	256.11 ± 6.86 ^D^
Magnesium	44.67 ± 5.80 ^A^	48.48 ± 4.61 ^A^	67.93 ± 0.89 ^B^	51.68 ± 4.43 ^A^	49.64 ± 2.79 ^A^	44.13 ± 4.10 ^A^	47.07 ± 6.96 ^A^
Calcium	9.38 ± 1.10 ^A^	6.32 ± 0.18 ^B^	16.28 ± 0.04 ^C^	11.77 ± 0.23 ^A^	6.58 ± 0.19 ^B^	7.40 ± 0.49 ^B^	6.50 ± 6.71 ^B^
Phosphorus	171.27 ± 6.38 ^A^	140.16 ± 3.99 ^B^	217.17 ± 1.45 ^C^	182.20 ± 55.53 ^A,C^	144.35 ± 34.78 ^A,B^	221.42 ± 8.66 ^C^	158.26 ± 17.47 ^A,B^
Manganese	9.30 ± 1.78 ^A^	8.05 ± 0.56 ^A^	9.02 ± 1.29 ^A^	6.46 ± 0.71 ^B^	14.17 ± 1.22 ^C^	3.29 ± 2.68 ^D^	1.68 ± 0.26 ^E^
Zinc	0.88 ± 0.12 ^A^	0.75 ± 0.08 ^A^	0.88 ± 0.15 ^A^	0.61 ± 0.13 ^A^	1.28 ± 0.64 ^A,B,C^	1.35 ± 0.09 ^B^	1.69 ± 0.06 ^C^
Copper	0.10 ± 0.02 ^A^	0.13 ± 0.01 ^A^	0.25 ± 0.00 ^B^	0.21 ± 0.11 ^A,B^	0.19 ± 0.00 ^A,B^	0.27 ± 0.01 ^B^	0.38 ± 0.00 ^C^

Data are expressed as mean ± standard deviation (g/100 g in wet-based). * n = 9 (3 batches, triplicate analysis); ** n = 15 (3 batches, quintuplicate analysis). Different uppercase letters on the same line indicate significant differences (Tukey’s test, *p* < 0.05). ACHO: available carbohydrates.

**Table 3 foods-14-00372-t003:** Fatty acid levels in plant-based burgers marketed in Brazil.

Fatty Acid	Plant-Based Burgers						
	SPGL	SPCHP	SOY1	SOY2	QUI	PLENT	CHP
	Mean ± SD	Mean ± SD	Mean ± SD	Mean ± SD	Mean ± SD	Main ± SD	Main ± SD
C6:0	-	1.44 ± 0.13	1.59 ± 0.15	1.86 ± 0.85	-	1.84 ± 1.20	-
C8:0	0.39 ± 0.17	21.76 ± 1.00	24.71 ± 1.39	11.51 ± 11.58	0.16 ± 0.05	26.20 ± 16.69	0.49 ± 0.10
C10:0	0.39 ± 0.15	17.57 ± 0.64	24.54 ± 1.36	12.95 ± 10.48	-	21.55 ± 13.50	-
C11:0	-	-	0.40 ± 0.05	-	-	0.30 ± 0.17	-
C12:0	3.20 ± 1.60	133.24 ± 4.70	326.28 ± 19.73	205.73 ± 37.82	0.51 ± 0.35	171.35 ± 95.59	3.50 ± 1.29
C13:0	-	-	0.38 ± 0.086	0.33 ± 0.05	-	0.25 ± 0.07	8.96 ± 4.57
C14:0	6.80 ± 1.84	51.58 ± 1.96	103.21 ± 6.35	75.64 ± 19.31	11.40 ± 0.49	65.01 ± 37.92	2.54 ± 0.73
C15:0	0.29 ± 0.02	0.50 ± 0.14	0.18 ± 0.025	0.31 ± 0.06	0.37 ± 0.03	0.34 ± 0.13	0.41 ± 0.23
C16:0	219.75 ± 20.49	65.55 ± 3.46	87.07 ± 2.05	106.90 ± 6.75	112.97 ± 3.69	62.98 ± 10.50	69.02 ± 3.50
Not identified	0.28 ± 0.06	-	-	-	-	0.19 ± 0.08	0.27 ± 0.08
C16:1n9	-	0.34 ± 0.07	-	0.18 ± 0.03	0.23 ± 0.04	-	0.61 ± 0.16
C16:1n7	2.90 ± 0.75	1.53 ± 0.10	0.34 ± 0.05	0.71 ± 0.16	1.27 ± 0.06	0.61 ± 0.08	-
Not identified	0.17 ± 0.07	-	-	0.80 ± 0.07	0.63 ± 0.09	0.56 ± 0.09	0.54 ± 0.22
C17:0	0.87 ± 006	0.45 ± 011	0.61 ± 0.06	-	-	0.49 ± 0.24	-
Not identified	-	-	0.36 ± 0.04	-	-	0.31 ± 0.14	0.41 ± 0.06
C17:1	-	-	-	0.28 ± 0.03	27.47 ± 1.14	32.72 ± 1.80	38.05 ± 3.10
C18:0	31.54 ± 1.52	24.36 ± 0.83	137.62 ± 4.86	35.16 ± 1.59	-	-	-
C18:1n9*t*	1.43 ± 0.27	1.28 ±0.16	1.00 ± 0.16	1.06 ± 0.20	-	1.11 ± 0.39	0.90 ± 0.39
C18:1n9	248.24 ± 39.03	439.48 ± 22.56	125.14 ± 7.25	215.68 ± 38.68	256.91 ± 6.50	444.07 ± 90.09	258.37 ± 14.02
C18:2n6*t*	2.65 ± 0.31	-	0.87 ± 0.08	1.26 ± 0.26	1.32 ± 0.01	0.34 ± 0.18	-
C18:2n6*t*	2.51 ± 0.36	-	0.81 ± 0.08	1.16 ± 0.22	0.81 ± 0.02	0.24 ± 0.13	-
C18:2n6	366.40 ± 39.49	117.24 ± 4.05	143.74 ± 11.54	205.82 ± 37.04	393.31 ± 11.96	82.20 ± 21.78	496.48 ± 25.27
C18:3n6	-	4.04 ± 0.16	-	1.05 ± 0.19	1.43 ± 0.05	-	-
Not identified	-	0.83 ± 0.02	-	1.01 ± 0.18	-	-	13.80 ± 2.37
C20:0	3.06 ± 0.44	0.56 ± 0.07	2.39 ± 0.04	2.56 ± 0.36	3.09 ± 0.07	2.30 ± 0.31	-
C21:0	0.82 ± 0.28	-	-	-	-	-	-
C18:3n3	8.73 ± 3.19	58.82 ± 2.21	1.36 ± 0.15	17.06 ± 2.26	41.89 ± 0.72	6.44 ± 1.64	6.25 ± 0.93
C20:2n6	1.96 ± 0.40	1.04 ± 0.06	-	0.47 ± 0.13	0.64 ±0.05	0.64 ± 0.42	-
C22:0	0.61 ± 0.05	2.20 ± 0.09	2.58 ± 0.19	2.53 ± 0.34	3.55 ± 0.44	5.22 ± 1.26	1.24 ± 0.33
C20:3n3	0.65 ± 0.19	-	-	-	3.04 ± 0.11	-	-
C22:1n6	-	0.90 ± 0.18	0.40 ± 0.20	0.90 ± 0.18	-	0.55 ± 0.23	-
Not identified	-	0.73 ± 0.44	0.46 ± 0.40	0.73 ± 0.44	-	1.14 ± 1.13	-
C22:2n6	-	0.32 ± 0.17	-	-	0.38 ± 0.02	0.55 ± 0.23	-
C24:0	1.20 ± 0.16	1.01 ± 0.06	1.03 ± 0.07	1.25 ± 0.20	1.58 ± 0.04	1.89 ± 0.38	-
C24:1n6	0.91 ± 0.22	0.82 ± 0.05	-	-	0.32 ± 0.03	-	2.27 ± 0.311
FA Total	905.75 ± 35.12	948.13 ± 55.62	989.03 ± 18.50	913.37 ± 23.22	870.03 ± 23.20	931.06 ± 65.48	897.60 ± 40.96

Data are expressed as mean ± standard deviation (mg/g of total fat). n = 9 (3 batches, triplicate analysis). FA: fatty acid.

**Table 4 foods-14-00372-t004:** Fatty acid quality profile in plant-based burgers marketed in Brazil and in meat burgers (literature data).

Quality Index	Plant-Based Burgers (mg/g of Total Fat)							
	SPGL	SPCHP	SOY1	SOY2	QUI	PLENT	CHP	Meat Burger
	Mean ± SD (%)	Mean ± SD (%)	Mean ± SD (%)	Mean ± SD (%)	Mean ± SD (%)	Mean ± SD (%)	Mean ± SD (%)	Mean ± SD (%)
Σ SAT	268.92 ± 77.33 ^A^ (29.69 ± 8.53)	323.80 ± 10.6 ^A^ (34.16 ± 0.59)	714.99 ± 33.06 ^B^ (71.80 ± 2.22)	457.12 ± 113.37 ^C^ (50.05 ± 12.47)	157.16 ± 8.89 ^D^ (18.06 ± 12.41)	481.75 ± 174.39 ^C^ (41.14 ± 16.64)	123.96 ± 5.54 ^D^ (13.81 ± 0.61)	47.0
Σ MUFA	251.40 ± 110.42 ^A^ (27.75 ± 12.19)	443.52 ± 22.42 ^B^ (46.76 ± 0.91)	126.14 ± 7.13 ^C^ (12.68 ± 0.92)	218.37 ± 61.43 ^A^ (23.91 ± 6.72)	262.70 ± 11.62 ^A^ (30.19 ± 6.72)	453.35 ± 78.09 ^B^ (42.60 ± 12.55)	262.08 ± 16.12 ^A^ (29.20 ± 1.79)	42.0
Σ PUFA	383.81 ± 81.70 ^A^ (42.37 ± 9.02)	178.24 ± 6.00 ^B^ (18.64 ± 0.61)	145.07 ± 11.62 ^C^ (15.11 ± 1.37)	214.84 ± 45.07 ^D^ (23.52 ± 4.93)	442.07 ± 20.10 ^A^ (50.81 ± 4.93)	90.17 ± 23.33 ^E^ (9.81 ± 3.29)	510.28 ± 21.74 ^F^ (56.94 ± 3.07)	11.0
Σ TRANS	6.42 ± 2.00 ^A^ (0.71 ± 0.20)	1.28 ± 0.16 ^B^ (0.14 ± 0.00)	2.74 ± 0.26 ^C^ (0.27 ± 0.02)	3.28 ± 1.41 ^A,C^ (0.35 ± 0.15)	3.11 ± 0.61 ^C^ (0.35 ± 0.07)	1.49 ± 0.48 ^B^ (0.14 ± 0.06)	0.89 ± 0.48 ^B^ (0.1 ± 0.05)	3.0
Σ ω 3	9.38 ± 3.38 ^A^ (1.16 ± 2.50)	58.82 ± 2.21 ^B^ (6.21 ± 0.08)	1.36 ± 0.15 ^C^ (0.14 ± 0.10)	17.60 ± 7.50 ^A^ (1.92 ± 0.82)	41.89 ± 1.39 ^D^ (1.92 ± 0.82)	6.38 ± 1.55 ^A^ (0.67 ± 0.22)	13.80 ± 2.37 ^A^ (1.53 ± 0.26)	0.5
Σ ω 6	374.43 ± 40.78 ^A^ (34.28 ± 2.84)	121.88 ± 3.90 ^B^ (13.01 ± 0.53)	143.74 ± 1.06 ^A^ (14.53 ± 0.08)	209.76 ± 40.74 ^C^ (23.04 ± 4.46)	395.12 ± 18.59 ^D^ (23.03 ± 4.46)	83.42 ± 21.74 ^E^ (9.07 ± 3.14)	496.48 ± 25.27 ^F^ (55.31 ± 2.81)	10.4
Σ ω6/Σ ω3	39.91 ± 1.13 ^A^	2.07 ± 0.03 ^B^	105.69 ± 7.07 ^C^	11.90 ± 5.4 ^D^	9.43 ± 13.37 ^D^	13.07 ± 1.06 ^D^	35.97 ± 10.66 ^A^	20.8
ΣPUFA/ΣSAT	1.42 ± 0.41 ^A^	0.55 ± 0.00 ^B^	0.21 ± 0.02 ^C^	1.96 ± 0.86 ^A^	0.36 ± 0.02 ^D^	0.16 ± 0.28 ^D,C^	3.80 ± 1.02 ^A^	0.23
AI	0.42 ± 0.09 ^A^	0.64 ± 0.02 ^B^	5.43 ± 0.51 ^C^	1.33 ± 0.83 ^D^	0.22 ± 0.03 ^E^	1.03 ± 0.61 ^D^	0.11 ± 0.01 ^E^	0.64 *
TI	0.74 ± 0.21 ^A^	0.31 ± 0.00 ^B^	4.23 ± 0.32 ^C^	1.16 ± 0.37 ^A^	0.33 ± 0.02 ^B^	0.71 ± 0.27 ^B^	0.26 ± 0.01 ^B^	1.87 *

Data are expressed as mean ± standard deviation (mg/g of total fat). n = 9 (3 batches, triplicate analysis). Σ SAT: sum of the saturated fatty acids; Σ MUFA: sum of monounsaturated fatty acids; Σ PUFA: sum of polyunsaturated fatty acids; Σ TRANS: sum of *trans* fatty acids; Σ ω 3: sum of omega 3 fatty acids; Σ ω 6: sum of omega 6 fatty acids; AI: atherogenic index; TI: thrombogenic index. * Calculated for comparison (the authors of the original work did not present this result in the article). Different uppercase letters on the same line indicate significant differences (Tukey’s test, *p* < 0.05).

**Table 5 foods-14-00372-t005:** Essential and non-essential amino acid profile in plant-based burgers marketed in Brazil.

Amino Acid	Plant-Based Burgers (mg/g of Protein)						
	SPGL	SPCHP	SOY1	SOY2	QUI	PLENT	CHP
	Mean ± SD	Mean ± SD	Mean ± SD	Mean ± SD	Mean ± SD	Mean ± SD	Mean ± SD
Essential							
Threonine	63.02 ± 10.22 ^A^	76.53 ± 10.54 ^A^	66.33 ± 3.99 ^A^	72.43 ± 5.17 ^A^	68.49 ± 8.97 ^A^	64.69 ± 8.87 ^A^	57.12 ± 4.33 ^A^
Methionine + Cysteine	15.37 ± 2.16 ^A^	14.54 ± 1.81 ^A^	13.29 ± 0.25 ^A^	14.43 ± 1.27 ^A^	29.45 ± 3.90 ^B^	11.99 ± 1.73 ^A^	17.65 ± 1.01 ^A^
Phenylalanine + Tyrosine	120.97 ± 7.35 ^A^	76.42 ± 10.25 ^B^	112.52 ± 2.10 ^A^	105.31 ± 4.27 ^A^	105.60 ± 10.06 ^A^	112.70 ± 9.34 ^A^	81.21 ± 5.40 ^B^
Histidine	24.21 ± 2.71 ^A^	34.54 ± 4.75 ^B^	16.68 ± 0.00 ^C^	23.36 ± 3.81 ^A^	30.42 ± 6.78 ^A^	31.20 ± 6.17 ^A^	38.15 ± 2.16 ^C^
Lysine	45.29 ± 8.03 ^A^	88.94 ± 5.17 ^B^	103.00 ± 1.19 ^C^	88.93 ± 5.96 ^B^	61.60 ± 11.17 ^A^	115.30 ± 12.81 ^C^	98.02 ± 7.70 ^B^
Valine	49.65 ± 1.57 ^A^	42.63 ± 2.78 ^B^	37.58 ± 6.47 ^B^	40.99 ± 2.49 ^B^	41.09 ± 6.36 ^B^	43.84 ± 3.10 ^B^	49.93 ± 5.13 ^A,B^
Isoleucine	52.98 ± 8.63 ^A^	42.95 ± 6.30 ^A^	35.43 ± 1.09 ^B^	36.27 ± 1.22 ^B^	33.96 ± 7.41 ^B^	42.18 ± 6.03 ^A^	31.67 ± 4.60 ^B^
Leucine	62.15 ± 4.12 ^A^	58.31 ± 2.51 ^A^	47.46 ± 4.64 ^B^	59.04 ± 2.28 ^A^	54.97 ± 3.50 ^B^	65.53 ± 2.64 ^C^	54.17 ± 3.40 ^A,B^
Tryptophan	8.30 ± 0.80 ^A^	9.02 ± 0.12 ^A^	6.50 ± 1.06 ^B^	6.94 ± 0.55 ^B^	7.77 ± 0.56 ^A,B^	8.96 ± 0.53 ^A^	8.19 ± 0.67 ^A^
ΣEAA	447.24 ± 18.30 ^A^	450.08 ± 15.31 ^A^	438.77 ± 7.21 ^A^	446.91 ± 3.16 ^A^	426.54 ± 8.02 ^A^	498.15 ± 8.58 ^B^	436.10 ± 6.21 ^A^
Non-essential							
Alanine	45.62 ± 8.89 ^A^	40.70 ± 1.30 ^A^	37.16 ± 6.98 ^A^	30.07 ± 0.81 ^B^	48.35 ± 0.78 ^A^	31.23 ± 1.75 ^B^	37.59 ± 1.30 ^A,B^
Glycine	50.19 ± 2.35 ^A^	19.78 ± 1.76 ^B^	19.84 ± 1.48 ^B^	49.58 ± 4.07 ^A^	25.28 ± 3.97 ^B^	50.96 ± 9.78 ^A^	26.60 ± 0.58 ^B^
Prolyne	32.54 ± 2.98 ^A^	20.91 ± 1.19 ^B^	15.43 ± 0.97 ^C^	20.40 ± 3.25 ^B,C^	25.88 ± 4.37 ^A,B^	14.67 ± 3.72 ^C^	26.01 ± 0.48 ^D^
Hydroxyproline	4.83 ± 0.68 ^A^	5.27 ± 0.33 ^A^	5.06 ± 0.22 ^A^	4.94 ± 0.20 ^A^	5.84 ± 0.6 ^A^	4.92 ± 0.69 ^A^	4.25 ± 0.12 ^A^
Serine	88.52 ± 12.22 ^A^	67.83 ± 1.38 ^B^	77.60 ± 3.52 ^A^	78.14 ± 8.58 ^A^	68.14 ± 4.04 ^B^	79.45 ± 6.04 ^A^	78.79 ± 4.76 ^A^
Glutamic acid	105.80 ± 5.68 ^A^	121.16 ± 6.77 ^B^	131.80 ± 5.77 ^B,C^	139.80 ± 4.79 ^C^	159.88 ± 31.34 ^B,C^	140.41 ± 14.02 ^B,C^	124.15 ± 10.18 ^B,C^
Arginine	99.63 ± 9.62 ^A^	79.81 ± 7.12 ^B^	96.40 ± 4.53 ^A^	88.49 ± 7.24 ^A,B^	102.84 ± 16.92 ^A,B^	107.63 ± 8.81 ^A^	77.61 ± 8.24 ^B^
Asparagine	3.28 ± 0.43 ^A^	2.98 ± 0.13 ^A^	2.63 ± 0.12 ^B^	3.80 ± 0.59 ^A^	2.87 ± 0.17 ^A,B^	5.69 ± 0.29 ^C^	2.95 ± 0.17 ^A^
Cystine	12.36 ± 2.30 ^A^	9.95 ± 2.23 ^A^	13.33 ± 0.80 ^A^	17.19 ± 3.06 ^A^	14.29 ± 2.49 ^A^	11.58 ± 0.71 ^A^	15.41 ± 8.55 ^A^
Glutamine	8.03 ± 1.33 ^A^	7.09 ± 0.38 ^B^	25.53 ± 0.90 ^C^	13.14 ± 1.65 ^D^	7.83 ± 0.77 ^A^	14.57 ± 0.34 ^D^	7.23 ± 0.55 ^A^
Aspartic acid	108.83 ± 8.67 ^A^	102.25 ± 8.92 ^A^	107.32 ± 0.73 ^A^	121.06 ± 5.58 ^A^	103.11 ± 8.77 ^A^	121.74 ± 4.19 ^A^	101.77 ± 9.36 ^A^
Σ NEAA	582.27 ± 42.71 ^A^	482.30 ± 8.94 ^B^	532.10 ± 9.44 ^A^	561.25 ± 28.75 ^A^	564.29 ± 44.44 ^A^	582.06 ± 25.68 ^A^	502.96 ± 20.69 ^A,B^
AA Total	1029.51 ± 61.01 ^A^	932.38 ± 24.25 ^B^	970.87 ± 16.65 ^A,B^	1008.16 ± 52.46 ^A,B^	990.83 ± 52.46 ^A,B^	1080.21 ± 34.26 ^A^	939.06 ± 27.11 ^B^

Data are expressed as mean ± standard deviation (mg/g of protein). n = 9 (3 batches, triplicate analysis). Σ EAA: sum of essential amino acids; ΣNEAA: sum of non-essential amino acids. Different uppercase letters on the same line indicate significant differences (Tukey’s test, *p* < 0.05).

**Table 6 foods-14-00372-t006:** Amino acid chemistry score (AACS) of plant-based burgers marketed in Brazil in relation to WHO/FAO amino acid score for adults.

Plant-Based Burgers							
mg/g of Protein (Sample)/mg/g (WHO/FAO)							
Amino Acids	SPGL	SPCHP	SOY1	SOY2	QUI	PLENT	CHP
Threonine	2.73 ± 0.44	3.32 ± 0.45	2.88 ± 0.17	3.14 ± 0.22	2.97 ± 0.39	2.81 ± 0.38	2.48 ± 0.19
Sulfur Amino Acids (Methionine + Cysteine)	0.69 ± 0.09 *	0.66 ± 0.08 *	0.60 ± 0.01 *	0.65 ± 0.05 *	1.33 ± 0.18	0.54 ± 0.08 *	0.80 ± 0.04 *
Aromatic Amino Acids (Phenylalanine + Tyrosine)	3.18 ± 0.19	2.01 ± 0.26	2.96 ± 0.05	2.77 ± 0.11	2.77 ± 0.26	2.96 ± 0.24	2.13 ± 0.14
Histidine	1.61 ± 0.18	2.30 ± 0.31	1.11 ± 0.00	1.55 ± 0.25	2.03 ± 0.45	2.08 ± 0.21	2.54 ± 0.14
Lysine	1.00 ± 0.18	1.97 ± 0.11	2.29 ± 0.02	1.97 ± 0.13	1.36 ± 0.24	2.56 ± 0.28	2.17 ± 0.17
Valine	1.27 ± 0.04	1.09 ± 0.07	0.96 ± 0.01 *	1.05 ± 0.06	1.05 ± 0.16	1.12 ± 0.08	1.28 ± 0.13
Isoleucine	1.76 ± 0.28	1.43 ± 0.21	1.18 ± 0.03	1.20 ± 0.040	1.13 ± 0.24	1.40 ± 0.20	1.05 ± 0.15
Leucine	1.05 ± 0.07	0.99 ± 0.04	0.80 ± 0.01 *	1.00 ± 0.04	0.93 ± 0.05 *	1.10 ± 0.04	0.92 ± 0.06 *
Tryptophan	1.38 ± 0.13	1.50 ± 0.02	1.08 ± 0.17	1.16 ± 0.09	1.29 ± 0.09	1.49 ± 0.08	1.36 ± 0.11

* AACS values below 1.00 are considered to correspond to limiting amino acids.

## Data Availability

The original contributions presented in this study are included in the article/Appendix A. Further inquiries can be directed to the corresponding author.

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
