# Peer review of "Plant-Based Burgers in the Spotlight: A Detailed Composition Evaluation and Comprehensive Discussion on Nutrient Adequacy"

_foods, 2025, doi:10.3390/foods14030372_

Round 1
Reviewer 1 Report
Comments and Suggestions for Authors
The manuscript presents an interesting and relevant study on the evaluation of the nutritional profile of commercial plant-based burgers available in Brazil. The exploration of proximate and mineral composition and its fatty acid and amino acid profiles furnish a detailed snapshot of what consumers can expect nutritionally from these products. However, several aspects of the manuscript require improvement before publication:
1、 Abbreviations like PBBs should be clearly explained when they are first used, and then used in the same way all the time after that. For example, the first appearance of PBBs in the abstract did not include the full name (line 15), while the full name of PBBs appeared several times in the introduction (lines 55 and 104).
2、 What are the criteria and principles for selecting samples? Were there any market share considerations, trends in consumer preference, or other factors that influenced this selection?
3、 A comparative analysis of the nutrient content of plant-based burgers and meat burgers was carried out. It's good, and it's also suggested to compare the results with the nutrient content standards for plant-based burgers. Do all burgers tested meet the nutritional standards for plant-based burgers?
4、 It's suggested to add an analysis of the difference in nutrient content between a plant-based burger and a typical meat burger.
5、 Can eating plant-based burgers satisfy people's need for a healthy diet?
6、 Because the nutritional composition is quite different, what possible health effects will there be for consumers who choose PBBs instead of meat burgers or the other way around?
Author Response
The manuscript presents an interesting and relevant study on the evaluation of the nutritional profile of commercial plant-based burgers available in Brazil. The exploration of proximate and mineral composition and its fatty acid and amino acid profiles furnish a detailed snapshot of what consumers can expect nutritionally from these products.
Response: We appreciate and agree with the reviewer’s comments.
However, several aspects of the manuscript require improvement before publication:
- 1. Abbreviations like PBBs should be clearly explained when they are first used, and then used in the same way all the time after that. For example, the first appearance of PBBs in the abstract did not include the full name (line 15), while the full name of PBBs appeared several times in the introduction (lines 55 and 104).
Response: The abbreviations has been revised.
- 2. What are the criteria and principles for selecting samples? Were there any market share considerations, trends in consumer preference, or other factors that influenced this selection?
Response: We thank the reviewer for their comment. First, we selected the most prevalent plant-based burgers in Brazilian markets, as these represent the products most widely consumed by the population. The list of brands was sourced from a previous study conducted by our research group (10.1016/j.crfs.2024.100796). From this list, we selected products featuring diverse protein sources, providing a comprehensive overview of the nutritional profiles of plant-based burgers available in Brazil. This information was added in the text (lines 120 – 122).
- 3. A comparative analysis of the nutrient content of plant-based burgers and meat burgers was carried out. It's good, and it's also suggested to compare the results with the nutrient content standards for plant-based burgers. Do all burgers tested meet the nutritional standards for plant-based burgers?
Response: We thank the reviewer for their question. In discussing the results, we highlighted the absence of regulations in Brazil standardizing the nutritional composition of plant-based burgers, which contributes to the significant variability observed across brands. This study aims to urge regulatory authorities to establish clear and effective legislation for these products to better safeguard consumers.
- 4. It's suggested to add an analysis of the difference in nutrient content between a plant-based burger and a typical meat burger.
Response: We thank the reviewer for their suggestion. The nutritional composition of selected PBBs was evaluated against the identity and quality standards for meat burgers, as defined by Brazilian legislation. Additionally, the lipid profiles of PBBs and meat burgers available in Brazil were compared using data from the literature. We believe these comparisons provide a solid basis for drawing meaningful relationships between the analyzed products.
- 5. Can eating plant-based burgers satisfy people's need for a healthy diet?
Response: Answering this question is quite complex. A healthy diet encompasses multiple factors beyond the nutritional composition of individual foods. It also involves food safety considerations, cultural diversity of diets, food cost, environmental sustainability throughout the production and consumption chain. Our approach focused on evaluating the nutritional composition, using specific markers—such as fatty acid health indicators—as examples. While we believe this study provides valuable insights into the healthiness of plant-based diets, it should not be regarded as the sole factor for assessing dietary healthfulness.
6. Because the nutritional composition is quite different, what possible health effects will there be for consumers who choose PBBs instead of meat burgers or the other way around?
Response: Thanks for the excellent question. This limitation is commonly observed in food consumption studies conducted worldwide. Further research is essential to better characterize plant-based foods and accurately evaluate the impact of these diets on population health. This topic is discussed in the introduction, specifically in lines 73 to 78.
Reviewer 2 Report
Comments and Suggestions for Authors
re: Review of a scientific article
article number: foods-3386909
title: Plant-Based Burgers in the Spotlight: A Detailed Composition Evaluation and Comprehensive Discussion on Nutrient Adequacy
Dear Authors,
below I have included the review of an article titled “Plant-Based Burgers in the Spotlight: A Detailed Composition Evaluation and Comprehensive Discussion on Nutrient Adequacy”.
General opinion
The article aligns well with the theme of the Special Issue of Foods magazine titled “The Study and Development of Plant-Based Alternatives to Animal Food Products”.
The article is very interesting and addresses the current problem related to the production of meat product analogues that are intended to replace meat in meatless diets.
The popularity of plant-based products imitating meat products has grown significantly over the past few years. A special category among them is convenience food products. In response to growing consumer demand, there is a clear expansion of the range of products from the convenience food category, which are supposed to imitate meat products and act as their substitutes in the diet. Meat analogues are made from various plant materials rich in protein. Vegetable oils are most often used as a source of fat. Producers, wanting to attract consumers, make every effort to ensure that their products imitate meat originals as much as possible. Meat analogues usually imitate the appearance of meat products quite well, while imitating other sensory features is more difficult to achieve. The question also arises to what extent both groups of food products: meat products and their meatless analogues are similar in terms of nutritional value. Therefore, it seems fully justified to attempt to assess the quality of selected burger assortments produced from plant-based ingredients in terms of their nutritional value and suitability for meeting nutritional requirements.
In this study, seven plant-based burgers (PBBs) available in retail trade on the Brazilian market were selected for comparative evaluation, and the selection criterion was the protein source used in the raw material composition (products based on different sources of plant protein were selected for the study). The scope of the analytical part of the work was very wide, because the quality assessment of PBBs included, among others, measurement of the content of basic chemical components and determination of the fatty acid profile and amino acid composition. The results obtained by the authors confirmed the assumption that the PBBs subjected to the evaluation differed in terms of the characteristics subjected to the evaluation, including the degree of meeting the human nutritional needs for nutrients. The promising results obtained in this study can be considered the fact that the PBBs were characterized by a fairly high nutritional value of protein, as they largely covered the demand for essential amino acids (EAA), if we assume the consumption of a given product in the amount of 100 g per day. The fat content of PBBs was relatively low, but differences in the fatty acid profile were found. According to the authors, one of the reasons for the differences in nutritional quality found between the tested products imitating meat burgers may be the lack of legal regulations that would provide guidelines for the production of this type of food products.
In my opinion, the study was properly planned and its objective was clearly defined.
The interpretations of results and study conclusions are supported by the data obtained.
The manuscript structure, flow and writing are clear and easy to understand.
The authors indicated the small number of PBB products evaluated as the main limitation of this work. After reviewing the article, I have only a few detailed comments listed below.
Detailed comments
1. Line 120: The authors state that the plant-based burgers used in the study „were purchased from local stores”. Please clarify whether these products were sold in packages, and if so, was there any nutritional information on the label?
2. Line 249: In the chapter titled “Results,” the authors present their research findings and compare them with those of other researchers. They also relate their findings to dietary recommendations and indicators of nutritional value. I suggest changing the chapter title to “Results and Discussion” for clarity.
3. Line 266-272: The authors report that some PBBs (QUI and CHP) had relatively high carbohydrate content. Do you know which ingredients used in the production of the aforementioned plant-based burgers were the source of the carbohydrates?
4. Line 339-347: The authors discuss the results regarding the iron content of PBBs. – Did the iron content in the products assessed result solely from its natural presence in the raw materials used in production? Was any of the products enriched with iron, e.g. the PLENT product, which exceeded the others in terms of the content of this element?
Recommendation
I recommend minor revisions.
Author Response
General opinion
The article aligns well with the theme of the Special Issue of Foods magazine titled “The Study and Development of Plant-Based Alternatives to Animal Food Products”.
The article is very interesting and addresses the current problem related to the production of meat product analogues that are intended to replace meat in meatless diets.
The popularity of plant-based products imitating meat products has grown significantly over the past few years. A special category among them is convenience food products. In response to growing consumer demand, there is a clear expansion of the range of products from the convenience food category, which are supposed to imitate meat products and act as their substitutes in the diet. Meat analogues are made from various plant materials rich in protein. Vegetable oils are most often used as a source of fat. Producers, wanting to attract consumers, make every effort to ensure that their products imitate meat originals as much as possible. Meat analogues usually imitate the appearance of meat products quite well, while imitating other sensory features is more difficult to achieve. The question also arises to what extent both groups of food products: meat products and their meatless analogues are similar in terms of nutritional value. Therefore, it seems fully justified to attempt to assess the quality of selected burger assortments produced from plant-based ingredients in terms of their nutritional value and suitability for meeting nutritional requirements.
In this study, seven plant-based burgers (PBBs) available in retail trade on the Brazilian market were selected for comparative evaluation, and the selection criterion was the protein source used in the raw material composition (products based on different sources of plant protein were selected for the study). The scope of the analytical part of the work was very wide, because the quality assessment of PBBs included, among others, measurement of the content of basic chemical components and determination of the fatty acid profile and amino acid composition. The results obtained by the authors confirmed the assumption that the PBBs subjected to the evaluation differed in terms of the characteristics subjected to the evaluation, including the degree of meeting the human nutritional needs for nutrients. The promising results obtained in this study can be considered the fact that the PBBs were characterized by a fairly high nutritional value of protein, as they largely covered the demand for essential amino acids (EAA), if we assume the consumption of a given product in the amount of 100 g per day. The fat content of PBBs was relatively low, but differences in the fatty acid profile were found. According to the authors, one of the reasons for the differences in nutritional quality found between the tested products imitating meat burgers may be the lack of legal regulations that would provide guidelines for the production of this type of food products.
In my opinion, the study was properly planned and its objective was clearly defined.
The interpretations of results and study conclusions are supported by the data obtained.
The manuscript structure, flow and writing are clear and easy to understand.
The authors indicated the small number of PBB products evaluated as the main limitation of this work. After reviewing the article, I have only a few detailed comments listed below.
Response: We appreciate the reviewer's thorough evaluation and are delighted that the manuscript was recognized for its scientific merit.
Detailed comments
- Line 120: The authors state that the plant-based burgers used in the study „were purchased from local stores”. Please clarify whether these products were sold in packages, and if so, was there any nutritional information on the label?
Response: The selection of the PBBs analyzed in this study was based on a prior investigation that evaluated the nutritional information on the labels of various PBBs available in Brazilian markets (10.1016/j.crfs.2024.100796). This information has been incorporated into the text to clarify the criteria used for selecting the analyzed products (lines 120 – 122).
- Line 249: In the chapter titled “Results,” the authors present their research findings and compare them with those of other researchers. They also relate their findings to dietary recommendations and indicators of nutritional value. I suggest changing the chapter title to “Results and Discussion” for clarity.
Response: Thanks for suggestion. This was done.
- Line 266-272: The authors report that some PBBs (QUI and CHP) had relatively high carbohydrate content. Do you know which ingredients used in the production of the aforementioned plant-based burgers were the source of the carbohydrates?
Response: Table 1 presents the main protein and fats ingredients of the analyzed PBBs. The QUI sample consists of white, red, and black quinoa, along with rolled oats. The CHP sample includes chickpeas, rolled oats, and rice flour. We believe that the presence of rolled oats and rice flour contributes to the high carbohydrate content of these products.
- Line 339-347: The authors discuss the results regarding the iron content of PBBs. – Did the iron content in the products assessed result solely from its natural presence in the raw materials used in production? Was any of the products enriched with iron, e.g. the PLENT product, which exceeded the others in terms of the content of this element?
Response: The analysis assessed the total iron (Fe) content in PBBs. However, only the PLENT sample specified fortification with iron and B-complex vitamins, including Thiamine (B1), Riboflavin (B2), Niacin (B3), Pyridoxine (B6), Folic Acid (B9), Cobalamin (B12), and Biotin (B7).
Recommendation
I recommend minor revisions.
Response: Thanks for suggestions.
